# Durability Investigation of Carbon Fiber Reinforced Concrete under Salt-Freeze Coupling Effect

**DOI:** 10.3390/ma14226856

**Published:** 2021-11-13

**Authors:** Yongcheng Ji, Wenchao Liu, Yanmin Jia, Wei Li

**Affiliations:** School of Civil Engineering, Northeast Forestry University, Harbin 150040, China; yongchengji@126.com (Y.J.); L3268929945@126.com (W.L.)

**Keywords:** CFRP reinforced concrete, durability, salt-freeze coupling effect, chloride ion permeation, mechanical properties, finite element analysis

## Abstract

In order to study the durability behavior of CFRP (carbon fiber reinforced polymer) reinforced concrete, three category specimens (plain, partially reinforced, and fully reinforced) were selected to investigate its performance variation concerning chlorine salt and salt-freeze coupled environment, which included the microscopic examination, the distribution of chloride ion concentration, and the compressive properties. By observing the microscopic of the specimens, the surface and cross-section corrosion deterioration was examined with increasing exposure time, and the physical behavior of CFRP and core concrete were discussed. The chloride ion diffusion test exerted that the chloride ion concentration in plain specimens is at least 200 times higher than that of fully reinforced specimens. Therefore, the effectiveness of CFRP reinforcement will be proved to effectively hinder the penetration of chloride ions into the core section. The formula of the time-dependent effect of concrete diffusivity with salt-freeze coupling effect was presented and its accuracy verified. A time-varying finite element model of chloride ion distribution was established by using ABAQUS software. It can be seen from the axial compression test that the strength loss rate of three categories of specimens was varied when subjected to the corrosion environment. Therefore, it is proved that CFRP reinforcement can effectively reduce the deterioration of the specimen’s mechanical properties caused by the exposure environment. The research results can provide technical reference for applying the CFRP strengthened concrete in a severe salt-freeze environment.

## 1. Introduction

In cold coastal areas, many concrete buildings are subjected to the coupling effect of the freeze–thaw cycle and chlorine salt immersion (after this is referred to as the salt-freeze effect). The salt-freeze effect environment accelerates the deterioration process of a concrete structure and dramatically reduces the bearing capacity and durability of concrete structure [1]. CFRP reinforcement technology can effectively improve concrete structures’ bearing capacity and durability and have significant environmental protection advantages. On the one hand, relevant research found that the mechanical properties of CFRP decreased very slowly and had good corrosion resistance under acid, alkali, and salt environments [2,3]. Therefore, sticking CFRP on the concrete surface can effectively prevent the damage of chemicals to the concrete. On the other hand, CFRP has a lightweight, soft texture and excellent mechanical properties. The high tensile strength and high tensile elastic modulus of CFRP are used to improve the strength of the original concrete structure. Firstly, CFRP reinforcement technology is an effective way to solve the damage problem of concrete structures. Secondly, most scholars only consider the durability of CFRP reinforced concrete structures in the salt-freeze environment. At the same time, few researchers focus on the durability of CFRP reinforced concrete structures in the salt-freeze environment. Therefore, it is necessary to study the durability of CFRP reinforced concrete structures in a salt-freeze environment.

Some scholars have studied the durability of bonding strength between CFRP and concrete under chloride salt immersion. It was found that failure modes of unsoaked specimens included cohesion, shear, and bonding failure. The failure mode of the soaked specimens is the failure of the bonding surface between CFRP and concrete. It shows that the bonding performance between CFRP and concrete decreases after being soaked in chloride salt. Finally, the reduction factor of bond strength between CFRP and concrete under a chloride salt immersion environment is proposed [4,5,6,7,8].

Other scholars have studied the diffusion law of chloride ions in concrete. It is found that the permeability of chloride ions in reinforced concrete beams increases with the increase in crack width. Reducing the water–cement ratio can inhibit the permeability of chloride ions in reinforced concrete beams [9,10]. Liu et al. found that compressive stress inhibits the diffusion of chloride ions in concrete, tensile stress promotes the diffusion of chloride ions in concrete, and carbonation can promote the diffusion of chloride ions [11]. Finally, a calculation model of chloride ion concentration considering the influence of carbonation is proposed, and the model is verified in combination with the experimental results. Cheng et al. found that compressive stress does not permanently inhibit the diffusion of chloride ions in concrete [12]. The permeability of chloride ions in concrete is the lowest when the lateral compressive stress is close to 0.15 times the compressive strength of concrete. Chen et al. proposed the diffusion model of chloride ions in reinforced concrete structures and proved the stability and convergence of the model [13].

Some research focuses on the tensile durability of epoxy adhesives in aqueous solution and chloride solution, respectively. The tensile properties of the two epoxy adhesives are better evaluated by the method based on the flexibility beam. It is found that the degradation of tensile properties of specimens in chloride solution is more severe than that in tap water solution [14,15]. Rudawska studied the compressive durability of epoxy adhesive in an aqueous solution and chloride solution, respectively [16]. It is found that the degradation of compressive properties of specimens immersed in chloride solution is more severe than that in tap water solution. Rudawska A et al. studied the durability of epoxy adhesive in an FeSO_4_ solution immersion environment [17]. It is found that the elastic modulus of the specimen decreases with the increase in drying time, and the compressive strength of the specimen increases with the increase in drying time. Therefore, an appropriate concentration of FeSO_4_ solution will improve the compressive properties of epoxy resin. Few researchers studied the durability of CFRP composites in a chloride environment. It is found that the tensile properties of CFRP composites decrease slightly in a chloride environment. The damage of CFRP composites in a seawater immersion environment is more severe than that in a distilled water immersion environment because the thermal expansion coefficient of carbon fibers is different from that of epoxy resin. In addition, the hydrolysis of an epoxy matrix can also lead to contact surface damage [18]. Hong et al. studied the durability of CFRP composites in aqueous, acid, and alkali solutions, respectively [19]. It is found that the tensile strength and elastic modulus of CFRP composites decrease slightly in the immersion solution, and the acid solution deteriorates the carbon fiber reinforced composites most seriously.

In this study, three category specimens (plain, partially reinforced, and fully rein-forced) were selected to investigate its performance variation concerning chlorine salt and salt-freeze coupled environment. Firstly, during the exposure process, the mass of the concrete specimen and the pH value of the immersion solution is measured. Secondly, the surface and cross-section of the specimen are observed with a microscope. Then, powder samples are taken at different locations to measure the chloride concentration, and a time-varying formula for chloride concentration is proposed. An ABAQUS diffusion model is established based on the presented equation. Thirdly, the axial compression test is carried out to discuss the influence of exposure environment on the mechanical properties of the specimens, and the stress characteristics of three types of specimens are studied by combining the ABAQUS compression model. Finally, the durability behavior of the CFRP strengthened concrete in a chlorine salt and salt-freeze coupled environment is discussed.

## 2. Experimental Programs

### 2.1. Materials

CFRP composite consists of a layer of carbon fiber woven fabric and epoxy adhesive on both sides in this test. Generally, there are one-way carbon fiber woven fabric and two-way carbon fiber woven fabric. The one-way carbon fiber woven fabric was used in this test, consisting of 12 K pure carbon fiber filaments. The single-layer thickness of carbon fiber woven fabric is 0.167 mm. The epoxy adhesive includes A and B two-component, in which component A is epoxy resin and B is the curing agent. The mixing mass ratio of A and B is 2:1. Therefore, the CFRP composite is a single-layer one-way plate. In order to obtain the tensile properties of CFRP composite and epoxy adhesive, the sheet specimens of epoxy adhesive and CFRP composite were prepared according to ASTM D3039. The sheet specimens of CFRP composite is a single-layer unidirectional plate, and the tensile test results are shown in Table 1. The constituent materials of concrete include water, cement, sand, and coarse aggregate. The mass ratio of each component is 0.54:1:1.64:3.12. The coarse aggregate is divided into two parts according to the particle size. The mass proportion of coarse aggregate with a 5–10 mm particle size is 25%, and the mass proportion of coarse aggregate with a 10–25 mm is 75%. The raw materials of concrete are shown in Figure 1.

### 2.2. Specimen Preparation

The plain, partially reinforced, and fully reinforced specimens were selected to discuss the effect of CFRP reinforcement area ratio on the specimen’s mechanical performance. The shape of the specimens is a cylinder with a diameter of 100 mm and a height of 200 mm. The CFRP bonding area ratios on the side of the specimen are 0%, 50%, and 100%, respectively. Two widths of CFRP composite are used, which includes 200 mm and 20 mm, respectively. The plain specimens are not reinforced. Instead, the partially reinforced specimen are reinforced with 5 CFRP composites with a width of 20 mm, and the CFRP composites are evenly spaced 25 mm along with the height of the concrete specimens. Similarly, fully reinforced specimens are reinforced with 1 CFRP composites with a width of 200 mm, which equals the same specimens’ height. Generally, there are two failure modes of CFRP composites on the specimen surface, namely material fracture and bonding failure at the interface. The former can give full play to the mechanical properties of CFRP composites. In order to ensure the reinforcement effect, the adhesive force at the interface must be strengthened, so the two ends of the CFRP composite need to be overlapped. According to relevant research, the overlap length is about 1/4 of the circumference of the specimen circular section to prevent debonding at the interface section. Firstly, polish the reinforced part of the concrete specimen to make it rough. Secondly, epoxy adhesive is applied on both sides of the carbon fiber woven fabric to prepare a CFRP composite. Then paste the CFRP composite on the reinforced part of the concrete specimen. Finally, leave the concrete specimens at room temperature for seven days until the epoxy adhesive solidifies. In addition, epoxy adhesive is applied on the top and bottom surfaces to retard the chloride penetration path for partially and fully reinforced specimens. The specimen size and reinforcement location are shown in Figure 2.

### 2.3. Exposure Environment

The chlorine salt freeze–thaw environment is set according to “The Standard of Testing Methods for Long-Term Performance and Durability of Common Concrete” (GB/T 50082-2009). A freeze–thaw cycle lasts for 4 h. The upper and lower limits of the center temperature of concrete specimens are 8 ± 2 °C and −17 ± 2 °C, respectively. The freeze–thaw medium is NaCl solution with a mass concentration of 3.5%, as shown in Figure 3a. In order to contrast with chlorine salt freeze–thaw environment, the chlorine salt immersion environment was set. The immersion medium is NaCl solution with a mass concentration of 3.5%, and the soaking temperature is room temperature (25 °C), as shown in Figure 3b.

The specimens were divided into 15 groups according to the difference of reinforcement type and exposure environment. There are six specimens in each group, a total of 90 specimens. Firstly, take out one of each group of specimens for cutting, then observe its micro morpHology with a microscope, and finally, measure the chloride ion concentration at different positions. Then, another five specimens in each group shall be subjected to an axial compression test. The specimens’ grouping and label information are shown in Table 2. For example, in “FRSCSFT-100”, the “FRS” represents a fully reinforced concrete specimen, the “CSFT” represents a salt-freeze environment, and the “100” represents 100 freeze–thaw cycles.

### 2.4. PH and Mass Measurement

The change of pH value of NaCl solution and the change of specimen mass are used to characterize the damage degree of the exposed environment to the specimen. For the specimen in the chlorine salt immersion environment, the NaCl solution’s pH value and the specimen’s mass shall be measured every 100 h. In addition, the NaCl solution’s pH value and the specimen’s mass shall be measured every 25 cycles for the specimen in a chlorine salt freeze–thaw environment. The measurement process is shown in Figure 4a,b.

### 2.5. Microstructure Observation

In order to visually reflect the degree of damage to the concrete specimen by the exposure environment, an electron microscope (Shenzhen Youshi Hongtu Technology Corporation, Shenzhen, China) was used to observe the cross-section and surface of the concrete specimen. Firstly, cut a cross-section at half the height of the concrete specimen. Secondly, rinse the concrete specimen with clean water and silence it until the concrete specimen is dry. Finally, adjust the microscope scope, and the optimal magnification factor of 500 times is selected to observe the apparent property of the specimen, as shown in Figure 5.

### 2.6. Chloride Ion Concentration Test

The chloride ion concentration at different positions was measured to study the diffusion law in the concrete specimen. Firstly, grind the sample powder every 10 mm along the radius of the cross-section. Six sampling points are selected, including point A, point B, point C, point D, and E and point F, as shown in Figure 6a. Secondly, put the sample powder in an oven, bake at 150 ± 5 °C for 2 h, and cool to room temperature. Then weigh 0.01 g of sample powder, dissolve into 100 mL of distilled water, and stand for 8 h. Finally, the chloride ion concentration is measured with a chloride ion content rapid tester, as shown in Figure 6b.

### 2.7. Mechanical Test

The loading process of the axial compression test consists of a mechanical control system and strain acquisition system. The former is responsible for controlling the loading rate and collecting load data. YAM-5000F electro-hydraulic servo hydraulic pressure testing machine (Shanghai Hualong Test Instruments Corporation, Shanghai, China) is used to conduct an axial pressure test on the specimen, and the loading speed is 0.5 MPa/s. The latter is responsible for collecting strain gauges data. A strain gauge is bonded at 1/2 height of the specimen to obtain the axial strain of the specimen. The loading process and acquisition process is shown in Figure 7.

## 3. Results and Discussion

### 3.1. PH Value Variation and Mass Loss

According to Figure 8a or Figure 9a, the initial pH value of NaCl solution is 7.03. With the increase in the immersion period, the pH value of NaCl solution of plain specimens will rise rapidly, and then the rising speed will slow down. After immersion for 400 h, the pH value reached 12.22, increased by 73.8%, and tended to be stable in the later stage of immersion. The change law of the pH value of NaCl solution of partially reinforced specimens is similar to that of plain specimens, which is only slightly smaller than the latter under the same immersion period. After immersion for 400 h, the pH value reached 12.12, increased by 72.4%. The pH value of the NaCl solution of fully reinforced specimens increased very slowly. After immersion for 400 h, the pH value reached 7.16, increased by 1.8%. According to Figure 8b or Figure 9b, the initial pH value of NaCl solution is 7.03. With the increase in the freeze–thaw cycle, the pH value of the NaCl solution of plain specimens will rise rapidly. After 25 freeze–thaw cycles, the pH value reached 12.66, increased by 80.1%, and stabilized. The change law of pH value of NaCl solution of partially reinforced specimens is similar to that of plain specimens. After 25 freeze–thaw cycles, the pH value reached 12.49, increased by 77.7%. The pH value of the NaCl solution of fully reinforced specimens increased very slowly. After 25 freeze–thaw cycles, the pH value reached 8.91, increased by 26.7%.
(1)$$\mathrm{Si}O_{3}^{2-}+H_{2}O\overset{\mathrm{Hydrolysis}}{⇄}\mathrm{HSi}O_{3}{}^{-}+OH^{-}$$
(2)$$\mathrm{HSi}O_{3}{}^{-}+H_{2}O\overset{\mathrm{Hydrolysis}}{⇄}H_{2}\mathrm{Si}O_{3}+OH^{-}$$

The main component of concrete is CaSiO_3_, which is the salt of strong alkali weak acid. On the one hand, SiO_3_^2−^ reacts with H^+^ to produce HSiO_3_^−^, as chemical formula (1). On the other hand, HSiO_3_^−^ reacts with H^+^ to produce H_2_SiO_3_, as chemical formula (2). During the immersion or freeze–thaw of a concrete specimen, OH^−^ will dissociate into the NaCl solution, resulting in its alkalinity. The contact area between NaCl solution and concrete determines the change rate of pH value. To a certain extent, the change in pH value can represent the damage of the exposed environment to concrete. The greater the pH value variation, the more severe damage of concrete is observed. The change rate in pH value from large to small is plain specimens, partially reinforced specimens, and fully reinforced specimens. It can be explained that CFRP composites effectively prevent NaCl solution from directly contacting the concrete and weaken the damage of the exposed environment to the concrete. In addition, the change rate of pH in the salt-freeze environment is more significant than that in the chlorine salt immersion environment for the specimens with the same reinforcement type. The micro-cracks are generated on the surface and core concrete, and the structure becomes loose. It can be proved that the NaCl solution is easier to penetrate the specimen and increase the contact area with concrete. It shows that the deterioration of the salt-freeze environment to the specimen is more severe than that of the chlorine salt immersion environment.

According to Figure 10a or Figure 11a, the mass of plain specimens will increase rapidly during chlorine salt immersion. After immersion for 200 h, the mass increased by 1.79% and was then stable. After a week of standing and drying, the final value of the mass is 1.26% higher than the initial value. The law of mass change of partially reinforced specimens is similar to that of plain specimens. The growth rate of partially reinforced specimens is generally slower than that of plain specimens. After immersion for 200 h, the mass increased by 1.49%. After a week of standing and drying, the final value of the mass is 1.02% higher than the initial value. The mass of fully reinforced specimens will increase slowly during chlorine salt immersion. After immersion for 200 h, the mass increased by 0.12%. After a week of standing and drying, the final value of the mass is 0.12% higher than the initial value.

According to Figure 10b or Figure 11b, the mass of plain specimens will decrease rapidly during chlorine salt freeze–thaw. After 100 freeze–thaw cycles, the mass decreased by 11.22%. After a week of standing and drying, the final value of the mass is 12.49% lower than the initial value. Similarly, the mass of partially reinforced specimens decreased by 2.95% after 100 freeze–thaw cycles. After a week of standing and drying, the final value of the mass is 3.29% lower than the initial value. On the other hand, the mass of fully reinforced specimens increases at a prolonged rate during chlorine salt freeze–thaw. After 100 freeze–thaw cycles, the mass increased by 0.18%. After a week of standing and drying, the final value of the mass is 0.05% higher than the initial value. 

In the chlorine salt immersion environment, the growth rate of specimen mass from large to small is plain specimens, partially reinforced specimens, and fully reinforced specimens. The NaCl solution will slowly penetrate the specimen, increasing the mass of the specimen, and the CFRP composite can effectively hinder the penetration of the NaCl solution into the specimen. After drying, the final value of the specimen mass is slightly higher than the initial value. It is because a small amount of NaCl crystals remain after the evaporation of water in the specimen.

In the chlorine salt freeze–thaw environment, the mass of plain specimens and partially reinforced specimens decreases, and the mass of plain specimens decreases faster. In contrast, the mass of fully reinforced specimens increases slowly. In the chlorine salt freeze–thaw environment, a large matrix falls off the concrete surface of plain specimens and partially reinforced specimens. The mass of dropped matrix is greater than that of the NaCl solution absorption, resulting in the decline in the mass of the specimen. The matrix at the CFRP composite position of partially reinforced specimens does not fall off, so the mass decline rate of partially reinforced specimens is smaller than that of plain specimens. Due to the wrapping of CFRP composite, the matrix of fully reinforced specimens did not fall off. As a result, a small amount of NaCl solution penetrated the interior, increasing the mass of fully reinforced specimens.

### 3.2. Micro Observation

#### 3.2.1. Cross-Section Micro Observation

According to Figure 12, there are no micro-cracks in the cross-sections of plain specimens, partially reinforced specimens, and fully reinforced specimens under no deterioration environment.

The micro-cracks appear in the cross-section of plain specimens, and there are no micro-cracks in the cross-section of partially reinforced specimens and fully reinforced specimens after immersion for 200 h, as shown in Figure 13. Thus, it shows that the plain specimens have been damaged, while the partially reinforced specimens and fully reinforced specimens have no noticeable damage.

Similarly, after immersion for 400 h, the micro-cracks in the cross-section of plain specimens further expand and extend. The micro-cracks appear in the cross-section of partially reinforced specimens, as shown in Figure 14. There are no micro-cracks in the cross-section of fully reinforced specimens. It shows that the damage of plain specimens is aggravated, the partially reinforced specimens are damaged, but the fully reinforced specimens are not obviously damaged.

According to Figure 15, after 50 freeze–thaw cycles, micro-cracks appear in the cross-sections of plain specimens and partially reinforced specimens. There are no micro-cracks in the cross-sections of fully reinforced specimens. It shows that plain and partially reinforced specimens have been damaged, while fully reinforced specimens have no apparent damage. 

Similarly, Figure 16 shows the micro-cracks in the cross-sections of plain specimens, and partially reinforced specimens further expand and extend after 100 freeze–thaw cycles. Again, there are no micro-cracks in the cross-sections of fully reinforced specimens. It shows that the plain and partially reinforced specimens are further deteriorated, while the fully reinforced specimens have no noticeable damage.

#### 3.2.2. Surface Micro Observation

According to Figure 17, there are no micro-cracks in the cross-sections of plain specimens, partially reinforced specimens, and fully reinforced specimens under no deterioration environment. Therefore, only cement mortar on the surface can be observed, and only carbon fiber filaments can be observed on the surface of the CFRP composite. 

According to Figure 18, after immersion for 200 h, micro-cracks appear in the cross-section of plain specimens and partially reinforced specimens, and the cement mortar on the concrete surface is covered with NaCl crystals. In addition, NaCl crystals appear on the surface of CFRP composites of partially reinforced specimens and fully reinforced specimens. 

According to Figure 19, after immersion for 400 h, the micro-cracks in the cross-sections of plain specimens and partially reinforced specimens further expand and extend, and the NaCl crystals on the concrete surface further increased. As a result, the NaCl crystals on the CFRP composite surface of partially reinforced specimens and fully reinforced specimens increased.

According to Figure 20, after 50 freeze–thaw cycles, the matrix on the concrete surface of plain specimens and partially reinforced specimens falls off, exposing shallow gravel, and micro-cracks appear on the surface of the specimen. In addition, NaCl crystals appear on the surface of CFRP composites of partially reinforced specimens and fully reinforced specimens.

According to Figure 21, the matrix on the concrete surface of plain specimens and partially reinforced specimens falls off more seriously after 100 freeze–thaw cycles. As a result, the micro-cracks further expand and extend, and the NaCl crystals on the surface of CFRP composites of partially reinforced specimens and fully reinforced specimens increased further.

After immersion in NaCl solution and drying, the crystallization pressure is generated due to the precipitation of NaCl crystals, resulting in micro-cracks in the specimen. From the appearance and development of microcracks, the damage of specimens from large to small is plain, partially reinforced, and fully reinforced specimen. CFRP composites hinder the penetration of NaCl solution into concrete, and its radial binding force inhibits the emergence and development of microcracks. It shows that the CFRP reinforcement method can effectively improve the durability of concrete in a chloride immersion environment. The more the amount of CFRP is, the stronger the protection effect on concrete specimens is.

During the freeze–thaw cycle, the frost heaving force caused by pore water crystallization leads to micro cracks. From the appearance and development of microcracks, the damage of specimens from large to small is plain, partially reinforced, and fully reinforced specimen. This is because the radial binding force of CFRP composites counteracts the frost heaving force to a certain extent and inhibits the emergence and development of microcracks. It shows that the CFRP reinforcement method can effectively improve the durability of concrete in a chlorine salt freeze–thaw environment. The more the amount of CFRP used, the more protection effect on concrete specimens observed. In addition, from the shape and development of microcracks, the damage of chlorine salt freeze–thaw is more severe than chlorine salt immersion. Related research discusses the durability of CFRP reinforced concrete specimens in an H_2_SO_4_ solution [20]. The microstructure deterioration process of plain concrete exerted a severer degree than CFPR strengthened concrete in an H_2_SO_4_ solution. Therefore, it can be explained that the CFRP partially retard the penetration of the H_2_SO_4_ solution into the core concrete, which is consistent with this study.

### 3.3. Chloride Penetration Analysis

#### 3.3.1. Chloride Ion Concentration Time-Varying Model

Many studies show that the diffusion characteristics of chloride ions in concrete comply with Fick’s second law. Furthermore, relevant studies show that the chloride ion diffusion coefficient decreases exponentially during the period of chloride salt [21]. Based on the relevant research and test results, this paper presents the attenuation Formula (3) of chloride ion diffusion coefficient in concrete with chloride salt immersion period. The value of *D*_28_ is 36.2 × 10^−12^ m^2^·s^−1^, and the value of *m* is 0.052.
(3)$$D_{t}=D_{28}\times {\left(28/t\right)}^{m}$$
where *D_t_* is chloride ion diffusion coefficient after *t* days (m^2^·s^−1^); *D*_28_ is chloride ion diffusion coefficient after 28 days (m^2^·s^−^^1^); *t* is immersion period (days); m is attenuation index of diffusion coefficient.

Relevant studies show that the chloride ion diffusion coefficient increases linearly with the number of freeze–thaw cycles [22]. Therefore, this paper proposes a Formula (4) for the increase of chloride ion diffusion coefficient in concrete with the number of freeze–thaw cycles based on relevant research and test results. The value of *D*_0_ is 36.2 × 10^−12^ m^2^·s^−1^, and the value of *β* is 7.25 × 10^−13^.
(4)$$D_{NFT}=D_{0}\times \left(1+\beta N\right)$$
where *D_NFT_* is chloride ion diffusion coefficient after *N* freeze–thaw cycle (m^2^·s^−1^); *D*_0_ is chloride ion initial diffusion coefficient (m^2^·s^−1^); *β* is increase coefficient of diffusion coefficient; *N* is the number of freeze–thaw cycles.

Combined with Fick’s second law, Formulas (3) and (4), a time-varying model of chloride ion concentration is proposed, which can be expressed by Formula (5).
(5)$$C\left(x,t\right)=C_{S}\cdot \left[1-erf\left(\frac{x}{2\sqrt{Dt}}\right)\right]$$
where *erf*(*u*) is Gauss’s error function. *C*(*x*,*t*) is chloride mass concentration in concrete (kg/m^3^); *C_S_* is chloride mass concentration of immersion solution (kg/m^3^); *x* is the diffusion depth (m); *t* is the exposure period (s); and *D* is the chloride ion diffusion coefficient (m^2^·s^−1^).

#### 3.3.2. Finite Element Analysis Model

In order to explore the time-varying effect of chloride ion diffusion coefficient of concrete in an exposure environment, a time-varying model of chloride ion distribution in concrete specimens was established based on Formula (5) and ABAQUS software. The mesh division of the model is shown in Figure 22, in which the red part is concrete, and the blue part is the CFRP composite. The element type of finite element model is DC3D8. The boundary condition is the concentration of the NaCl solution, and the solubility and diffusion coefficient of the material is set. The solubility of both materials is 1. The variation formula of chloride ion diffusion coefficient with exposure time is proposed in Section 3.3.1.

According to the finite element model, the chloride ion concentration distribution nepHogram of PSSI-200, PSSI-400, PSSF-50, PSSF-100, FRSSI-200, FRSSI-400, FRSSF50, and FRSSF-100 is obtained, as shown in Figure 23 and Figure 24. According to the color distribution characteristics of the nepogram, the concentration in the red edge area is the highest, the concentration in the blue center area is the lowest, and the color transitions from the red edge to the blue center. Furthermore, the longer the exposure period, the larger the red area and the smaller the blue area, indicating that the chloride ion concentration increases with the exposure time.

The test and simulation values of chloride ion concentration of plain specimens in chlorine salt immersion and freeze–thaw environments are shown in Figure 25a,b, respectively. The test and simulated values of chloride ion concentration of fully reinforced specimens in chlorine salt immersion and freeze–thaw environments are shown in Figure 26a,b, respectively. The maximum error between the test and simulation values is only 8.7%, and the average error is only 5.1%, and the variation law of the two curves is very similar. This shows that the diffusion law of chloride ion in concrete specimens conforms to Fick’s second law and verifies the correctness of the model.

According to Figure 25a or Figure 26a, the chloride ion at each position of the plain concrete specimen is about 250 times that of the fully reinforced concrete specimen under the same immersion period. Similarly, Figure 25b or Figure 26b show that the chloride ion at each position of the plain concrete specimen is about 200 times that of the fully reinforced concrete specimen under the same freeze–thaw cycles. Thus, under the same exposure period, the chloride ion concentration at each position of the plain concrete specimen is much higher than that of the fully reinforced concrete specimen, indicating that CFRP reinforcement can effectively hinder the diffusion of chloride ions to the specimen.

All these observations can be explained by the many micro-cracks that occur on the surface and inside the specimen. As a result, the structure becomes loose, which accelerates the penetration rate of chloride ions to the specimen in the salt-freeze environment. However, concrete undergoes continuous hydration and its internal structure becomes compact, which hinders the penetration of chloride ions into concrete in the chlorine salt immersion environment. Related studies have established the chloride ion diffusion model of plain concrete specimens in the chloride salt immersion environment and the chloride salt freeze–thaw environment and discussed the diffusion law of chloride ions in the concrete [23]. Therefore, it can be concluded that the diffusion law of chloride ions in concrete satisfies Fick’s second law. This study established a chloride ion diffusion model for CFRP reinforced concrete specimens based on these studies. Furthermore, the model calibration is verified, indicating that the model can accurately predict CFRP reinforced concrete specimens’ chloride ion concentration.

### 3.4. Mechanical Performance Analysis

The failure modes of plain specimens under different exposure environments are the same. The specimen has no noticeable damage at the initial stage of loading. However, tiny cracks appear on the surface of the specimen with the progress of loading. Then small cracks begin to increase, expand, extend and cross each other, and finally develop into multiple discontinuous longitudinal cracks. The failure mode of the specimen is shown in Figure 27a.

The failure modes of partially reinforced specimens under different exposure environments are the same. At the initial stage of loading, the specimen has no noticeable damage. With the progress of loading, small cracks appear in the unreinforced part of the specimen, and the crack development is relatively slow. The reason is that CFRP composites inhibit the development of cracks. When the load reaches 80% of the ultimate bearing capacity of the specimen, the specimen makes a “crackling” sound, and partially epoxy adhesives crack. When the load reaches the ultimate bearing capacity of the specimen, the specimen will make a large “pop” sound. Three CFRP composites in the middle of the specimen broke instantaneously, accompanied by fragments of concrete splashing. The specimen shows apparent brittle failure, and the failure mode of the specimen is shown in Figure 27b. The ultimate bearing capacity of partially reinforced specimens is about 58% higher than that of plain specimens.

The failure modes of fully reinforced specimens under different exposure environments are the same. The specimen has no noticeable damage at the initial stage of loading. The specimen makes a “crackling” sound and partially epoxy adhesives crack as the load reaches 85% of the ultimate bearing capacity of the specimen. The specimen will make a large “pop” sound when the load reaches the ultimate bearing capacity of the specimen. The middle part of the CFRP composite broke instantaneously, accompanied by the splashing of concrete fragments. The specimen shows apparent brittle failure, and the failure mode of the specimen is shown in Figure 27c. The ultimate bearing capacity of fully reinforced specimens is about 148.2% higher than that of plain specimens.

The axial compression process of PSND, PRSND, and FRSND was simulated by the finite element software ABAQUS. Material properties of concrete are obtained from the tests, as shown in Table 3. The C3D8R element is selected to simulate concrete material [24,25,26]. The material properties of the CFRP composite are shown in Table 1. The thickness of CFRP composite is far less than the dimensions in other directions, and only its in-plane mechanical properties are considered in the finite element analysis. Therefore, the S4R element is selected to simulate the CFRP composite [27]. The hoop deformation between concrete and CFRP composite is coordinated during the loading process, and there is no relative with displacement. The contact property between concrete and CFRP composite cannot affect the compressive performance of concrete specimens but can only play a role in fixing the position of CFRP composite. Based on the above factors, the contact property between concrete and CFRP composite is set as binding [28,29]. The six degrees of freedom are completely restrained from fixing the lower specimen surface. A reference point RP1 is established to facilitate loading. A coupling relationship is set between the upper surface of the concrete specimen and the reference point RP1. The test loading rate is 0.5 MPa/s, which is converted into a concentrated load. The rate of concentrated load is 3.925 KN/s, which is applied to the reference point RP1.

The stress nepHogram of PSND is shown in Figure 28a. The random color distribution indicates that the stress distribution of the specimen is uneven and irregular. The model’s failure mode has a good agreement with the experimental test, as shown in Figure 27a or Figure 28a. In addition, the simulated compressive strength of the PSND specimen is 28.36 MPa, while the experimental compressive strength is 28.6 MPa, and the error between the two is only 0.84%.

The stress nepHogram of PRSND is shown in Figure 28b. Green represents the area with significant stress, mainly distributed in the position strengthened by CFRP composite, indicating that the stress of concrete is relatively large in the position reinforced by CFRP composite. Blue represents the area with minor stress, which is mainly distributed in the unreinforced position of the specimen. It can be concluded that the CFRP composite has a restraining effect on the concrete, which changes the concrete from a one-dimensional stress state to a three-dimensional stress state and can improve the compressive performance of the concrete. The model’s failure mode has a good agreement with the experimental test, as shown in Figure 27b or Figure 28b. In addition, the simulated compressive strength of the PRSND specimen is 45.02 MPa, while the experimental compressive strength is 45.2 MPa, and the error between the two is only 0.4%.

The stress nepHogram of FRSND is shown in Figure 28c. Green represents the area with enormous stress distributed in the whole model, indicating that the whole model is in a three-dimensional stress state. Blue represents the area with minor stress, which is the mechanical degradation area of the specimen after compression failure. There is blue in the middle of the specimen, indicating that the concrete damage in the middle is extensive. In addition, the simulated compressive strength of the FRSND specimen is 71.79 MPa, while the experimental compressive strength is 71.0 MPa, and the error between the two is only 1.11%. All these observations indicate that the three simulation effects have a good agreement with the test result.

According to Figure 28d, the stress of three CFRP composites in the middle of PRSND reaches 3520 MPa. Thus, the three CFRP composites in the middle of the specimen have reached the ultimate stress and are in the critical state of fracture, which is consistent with the failure mode of the test.

Similarly, Figure 28e shows that the stress in the middle of the carbon fiber composite of FRSND reaches 3520 MPa. Thus, the CFRP composite has reached the ultimate tensile stress and is in the critical state of fracture, which is consistent with the failure mode of the test.

Figure 29 and Figure 30 show the average ultimate compressive strength and strain of 15 groups of specimens. Table 4 shows the rate of change of ultimate compressive strength and ultimate strain for all specimens. Figure 31 shows the stress–strain relationship among specimens. The ultimate compressive strength of PSND is 28.6 MPa, and the ultimate strain is 4276.3 × 10^−6^. After 400 h chlorine salt immersion, the ultimate compressive strength decreases by 5.03%, the ultimate strain decreases by 16.69%, and the initial slope of stress–strain curves increase slightly. The ultimate compressive strength of PRSND is 45.2 MPa, and the ultimate strain is 10,422.5 × 10^−6^. After 400 h chlorine salt immersion, the ultimate compressive strength decreases by 3.94%, the ultimate strain decreases by 12.68%, and the initial slope of stress–strain curves increases slightly. The ultimate compressive strength of FRSND is 71 MPa, and the ultimate strain is 24,866 × 10^−6^. After 400 h chlorine salt immersion, the ultimate compressive strength decreases by 2.65%, the ultimate strain decreases by 4.31%, and the initial slope of the stress–strain curves is almost unchanged.

It is concluded that after chloride salt immersion, the ultimate compressive strength, ductility, and elastic modulus of plain specimens and partially reinforced specimens decrease slightly. NaCl crystals fill the pore in the specimen after chloride salt immersion, the overall structure is compact, and the elastic modulus increases slightly. During the process of chloride salt immersion, micro-cracks appear successively on the surface and in the interior. The micro-cracks of the specimens extend continuously during the compression process, which leads to the early destruction of the concrete structure and reduction of the bearing capacity of the specimens. After chloride salt immersion, the ultimate compressive strength and ductility of fully reinforced concrete specimens are reduced slightly, the modulus of elasticity is almost unchanged. This is because the CFRP composite prevents most NaCl solutions from penetrating the inside of the specimens during chloride salt immersion. Only a small amount of NaCl solution penetrates the inside of the specimens, which results in little change in the mechanical properties of fully reinforced concrete specimens. The mechanical properties of specimens change from large to small in order: plain specimens, partially reinforced specimens, and fully reinforced specimens. The larger the area of CFRP reinforcement, the smaller the influence of chloride salt immersion environment on the mechanical properties of specimens. A similar conclusion for the durability of CFRP reinforced concrete specimens in Na_2_SO_4_ solution is obtained [30]. The results show that plain and CFRP strengthened concrete specimens’ compressive strength and ductility decrease, and the elastic modulus increases subjected to environmental effect. However, the mechanical performance deterioration rate of CFRP reinforced concrete is lower than that of plain concrete specimens, indicating that the effectiveness of CFRP is confirmed.

The ultimate compressive strength of PSND is 28.6 MPa, and the ultimate strain is 4276.3 × 10^−6^. After 100 freeze–thaw cycles, the ultimate compressive strength decreases by 50.56%, the ultimate strain increases by 20.96%, and the initial slope of stress–strain curves decrease significantly. The ultimate compressive strength of PRSND is 45.2 MPa, and the ultimate strain is 10,422.5 × 10^−6^. After 100 freeze–thaw cycles, the ultimate compressive strength decreases by 16.12%, the ultimate strain increases by 12.63%, and the initial slope of stress–strain decreases significantly. The ultimate compressive strength of FRSND is 71 MPa, and the ultimate strain is 24,866 × 10^−6^. After 100 freeze–thaw cycles, the ultimate compressive strength decreases by 10.47%, the ultimate strain increases by 4.72%, and the initial slope of stress–strain decreases significantly.

Related studies discussed the durability of plain concrete in a chlorine salt freeze–thaw environment [31]. However, without fiber strengthened effect, the specimens’ strength and elastic modulus decrease rapidly in a chlorine salt freeze–thaw environment, and the durability of CFRP reinforced concrete needs to be further investigated. Furthermore, the mechanical behavior of CFRP reinforced concrete is effectively improved, such as compressive strength, elastic modulus, and ductility [32,33,34]. Thus, the performance deterioration rate between plain and CFRP reinforced concrete is discussed under environmental effects, and the effectiveness of various reinforcement ratios is qualified in this study. After chlorine salt freeze–thaw, the ultimate compressive strength and elastic modulus of plain, partially reinforced, and fully reinforced specimens, their ductility increases. This is because pore water crystallizes during the process of chlorine salt freeze–thaw and produces frost heaving force. Under the repeated frost heaving force, the matrix on the concrete surface falls off, the overall structure becomes crisper and crisper, and the elastic modulus becomes small. At the same time, micro-cracks appear on the surface and inside of the concrete. The micro-cracks of the specimen expand and extend during the compression process, resulting in the early failure of the concrete structure and the decreases of the bearing capacity. In the process of chlorine salt freeze–thaw, the circumferential binding force of CFRP composites offsets the frost heaving force, the internal structure is less affected, and there are no obvious micro cracks. Therefore, the mechanical properties of fully reinforced specimens change little. The mechanical properties of specimens change from large to small in order: plain specimens, partially reinforced specimens, and fully reinforced specimens. The larger the area of CFRP reinforcement, the smaller the influence of chlorine salt freeze–thaw environment on the mechanical properties of specimens.

## 4. Conclusions

This paper discusses the influence of CFRP reinforcement on the durability of concrete specimens from three aspects: micromorphology, chloride ion distribution characteristics, and mechanical properties. The specific conclusions are as follows.

From the microscopic observation results, the damage degree of the specimens is plain specimens, partially reinforced specimens, and fully reinforced specimens from large to small. It shows that the CFRP reinforcement method can effectively improve the durability of concrete in chlorine salt immersion or chlorine salt freeze–thaw environment. The more CFRP is used, the higher the durability of concrete specimens. In addition, from the development of microcracks and matrix falling state, the damage of chlorine salt freeze–thaw is more severe than chlorine salt immersion.

Under the same exposure period, the chloride ion concentration in plain specimens is at least 200 times higher than that of fully reinforced specimens. It indicates that CFRP reinforcement can effectively hinder chloride ion diffusion to the specimen. Under the same exposure period, the chloride ion concentration of the specimen in the chlorine salt freeze–thaw environment is more significant than that in the chlorine salt immersion environment. It indicates that the freeze–thaw cycle can promote the diffusion of chloride ions to the specimen. In addition, a time-varying formula for chloride ion concentration was proposed, and an ABAQUS model was established based on the formula.

According to the axial compression test, after 100 freeze–thaw cycles, the strength reduction rate of plain specimens is 50.56%, and that of fully reinforced specimens is 10.47%. It proves that CFRP reinforcement can effectively reduce the influence of exposure environment on the mechanical properties of specimens. In addition, compared with the chlorine salt immersion environment, the strength reduction rate of the specimen in the chloride freeze–thaw environment is greater.

## Figures and Tables

**Figure 1 materials-14-06856-f001:**
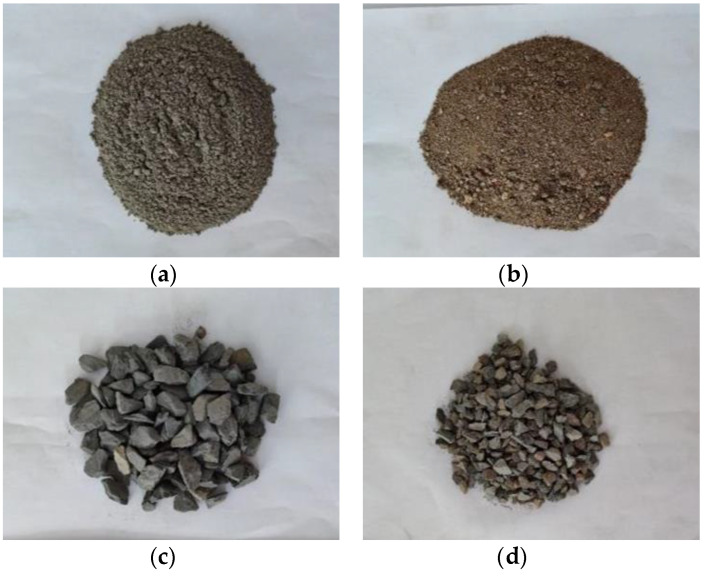
Raw material of concrete: (**a**) cement; (**b**) sand; (**c**) 10–20 mm coarse aggregate; (**d**) 5–10 mm coarse aggregate.

**Figure 2 materials-14-06856-f002:**
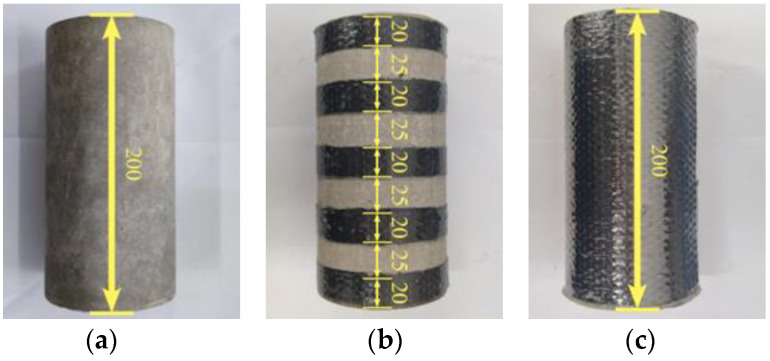
Concrete specimens (unit: mm): (**a**) plain specimen; (**b**) partially reinforced specimen; (**c**) fully reinforced specimen.

**Figure 3 materials-14-06856-f003:**
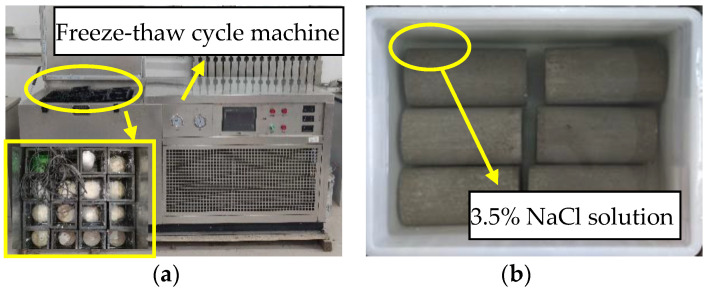
Exposure environment: (**a**) chlorine salt freeze–thaw environment; (**b**) chlorine salt immersion environment.

**Figure 4 materials-14-06856-f004:**
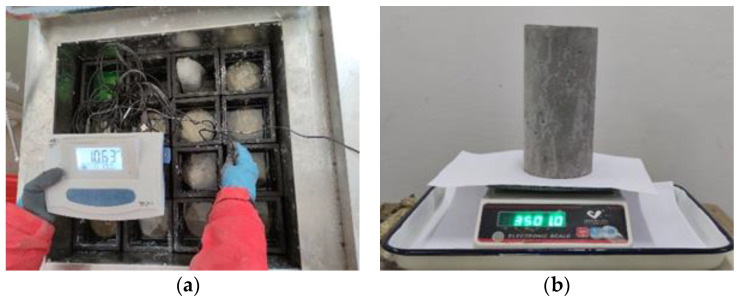
pH and mass measurement: (**a**) pH measurement; (**b**) mass measurement.

**Figure 5 materials-14-06856-f005:**
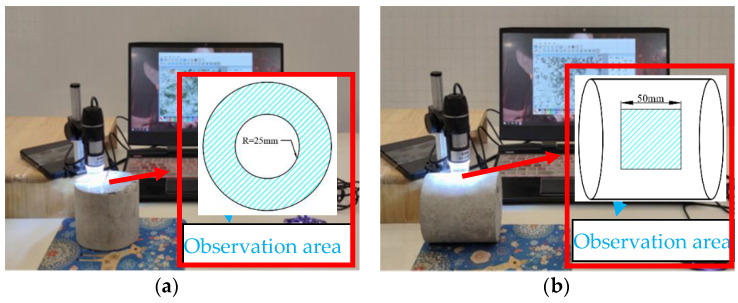
Observe the microscopic of the specimen (unit: mm): (**a**) cross-section observation; (**b**) surface observation.

**Figure 6 materials-14-06856-f006:**
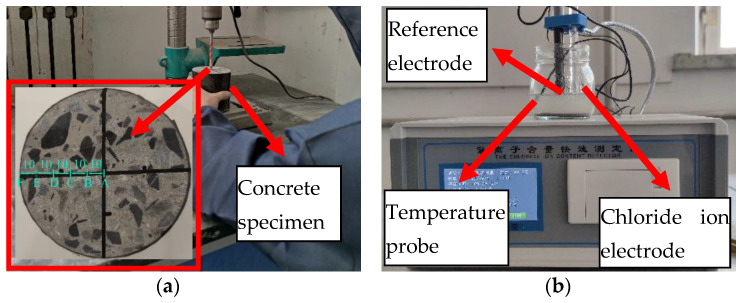
Sampling and measurement of chloride ion concentration (unit: mm): (**a**) sampling location; (**b**) chloride ion concentration measurement.

**Figure 7 materials-14-06856-f007:**
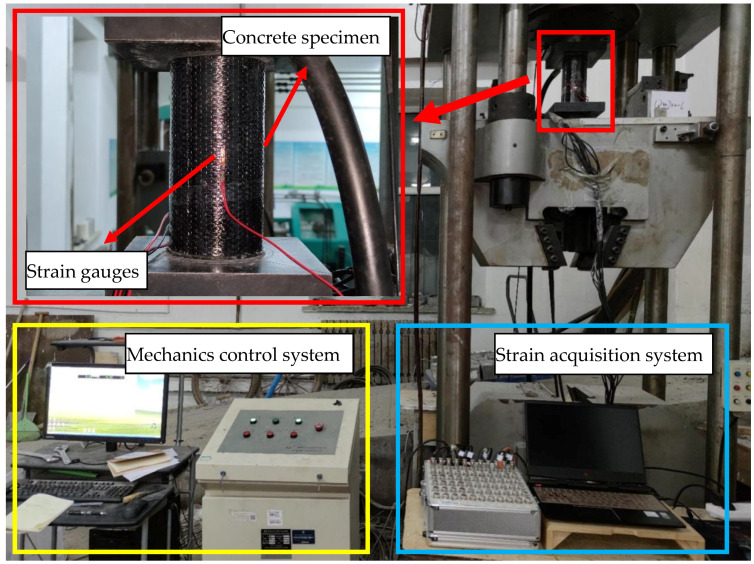
Mechanical test set up.

**Figure 8 materials-14-06856-f008:**
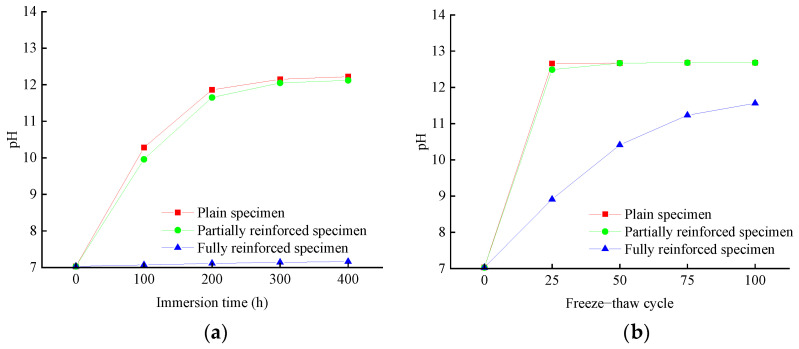
pH change of NaCl solution: (**a**) chlorine salt immersion environment; (**b**) chlorine salt freeze–thaw environment.

**Figure 9 materials-14-06856-f009:**
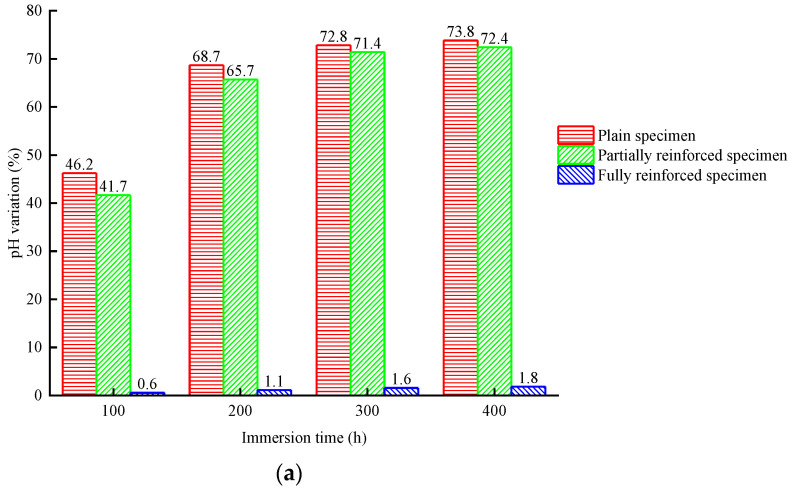
pH change rate of NaCl solution: (**a**) chlorine salt immersion environment; (**b**) chlorine salt freeze–thaw environment.

**Figure 10 materials-14-06856-f010:**
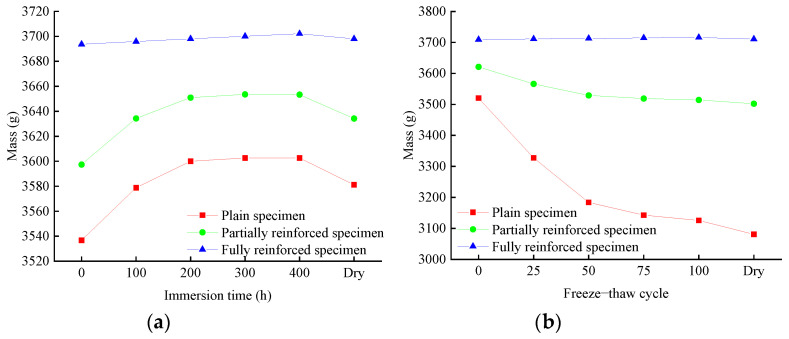
Mass change of specimen: (**a**) chlorine salt immersion environment; (**b**) chlorine salt freeze–thaw environment.

**Figure 11 materials-14-06856-f011:**
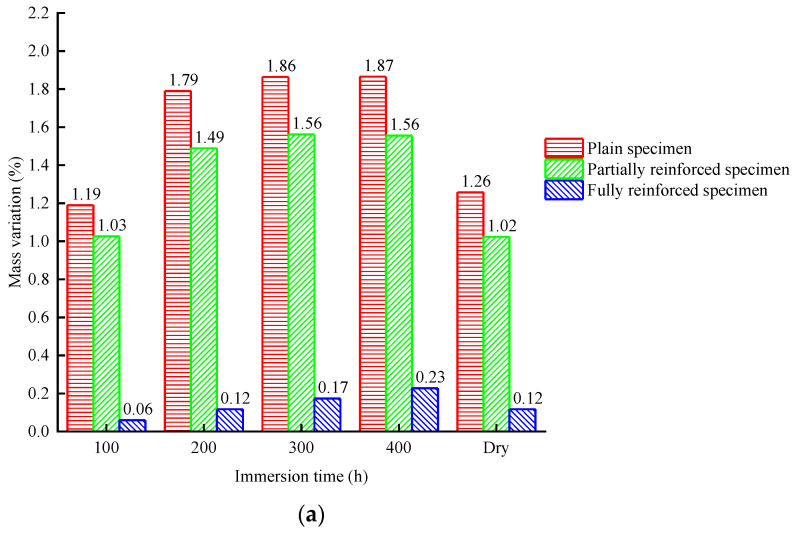
Mass change rate of specimen: (**a**) chlorine salt immersion environment; (**b**) chlorine salt freeze–thaw environment.

**Figure 12 materials-14-06856-f012:**
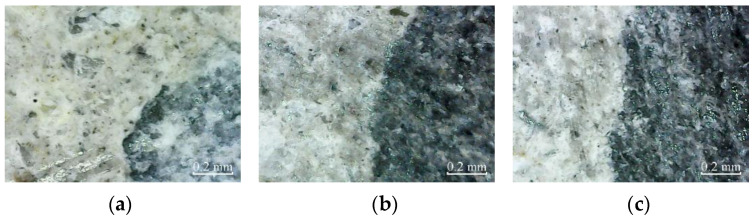
Cross-section: (**a**) PSND; (**b**) PRSND; (**c**) FRSND.

**Figure 13 materials-14-06856-f013:**
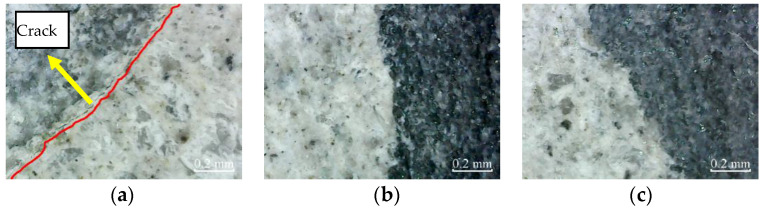
Cross-section: (**a**) PSCSI-200; (**b**) PRSCSI-200; (**c**) FRSCSI-200.

**Figure 14 materials-14-06856-f014:**
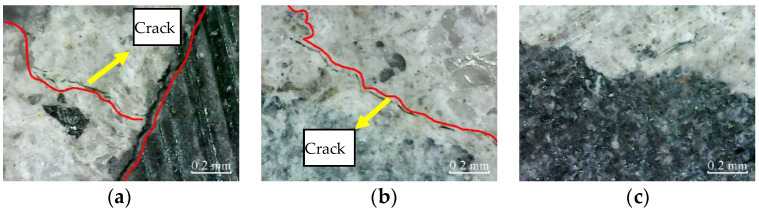
Cross-section: (**a**) PSCSI-400; (**b**) PRSCSI-400; (**c**) FRSCSI-400.

**Figure 15 materials-14-06856-f015:**
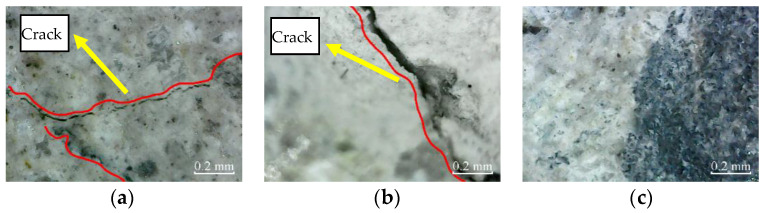
Cross-section: (**a**) PSCSFT-50; (**b**) PRSCSFT-50; (**c**) FRSCSFT-50.

**Figure 16 materials-14-06856-f016:**
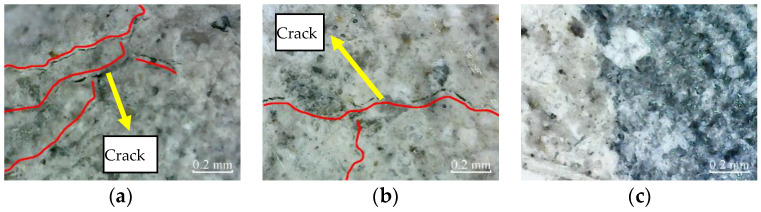
Cross-section: (**a**) PSCSFT-100; (**b**) PRSCSFT-100; (**c**) FRSCSFT-100.

**Figure 17 materials-14-06856-f017:**
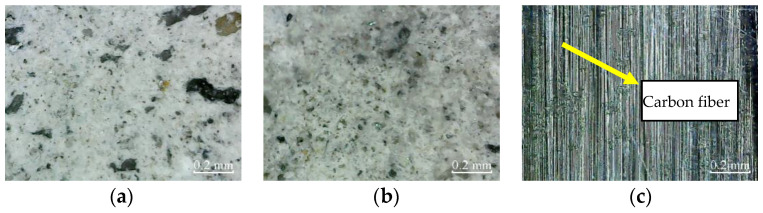
Cross-section: (**a**) PSND; (**b**) PRSND; (**c**) FRSND.

**Figure 18 materials-14-06856-f018:**
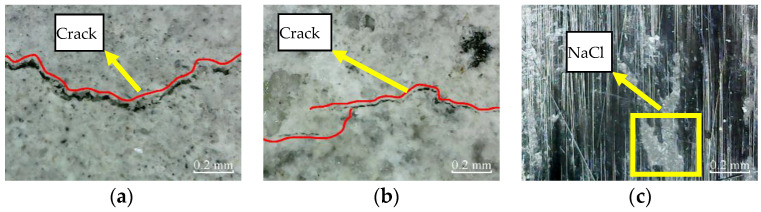
Cross-section: (**a**) PSCSI-200; (**b**) PRSCSI-200; (**c**) FRSCSI-200.

**Figure 19 materials-14-06856-f019:**
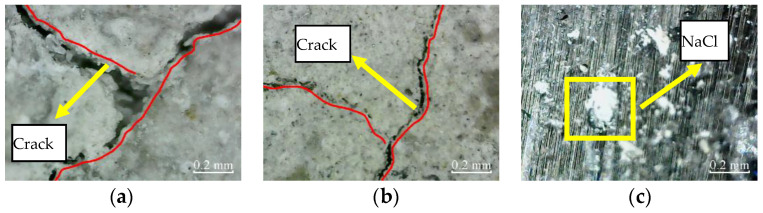
Cross-section: (**a**) PSCSI-400; (**b**) PRSCSI-400; (**c**) FRSCSI-400.

**Figure 20 materials-14-06856-f020:**
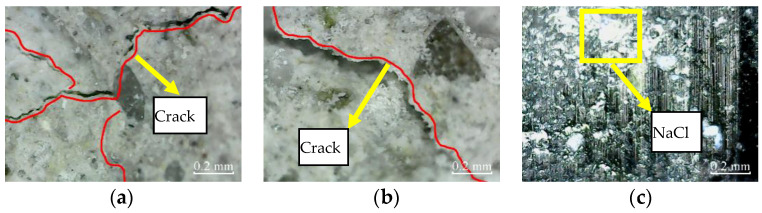
Cross-section: (**a**) PSCSFT-50; (**b**) PRSCSFT-50; (**c**) FRSCSFT-50.

**Figure 21 materials-14-06856-f021:**
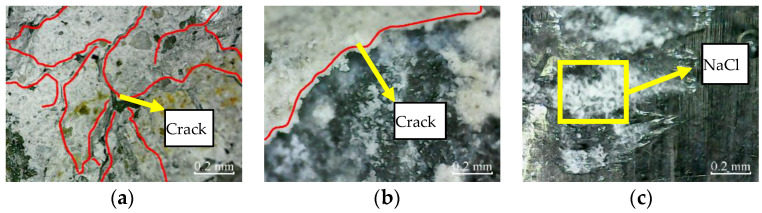
Cross-section: (**a**) PSCSFT-100; (**b**) PRSCSFT-100; (**c**) FRSCSFT-100.

**Figure 22 materials-14-06856-f022:**
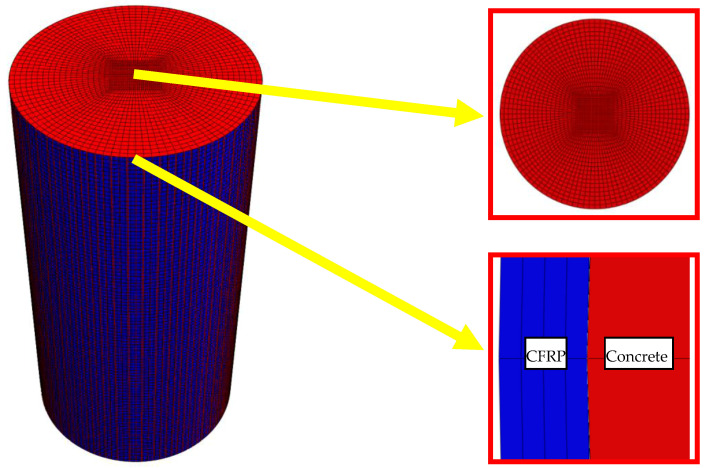
Mesh generation of finite element model.

**Figure 23 materials-14-06856-f023:**
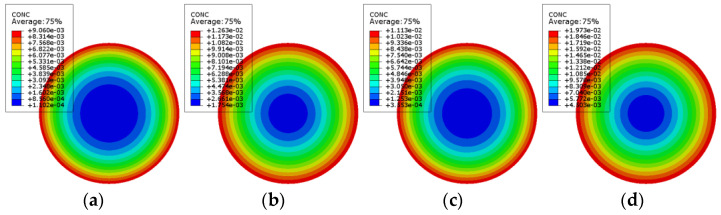
Plain concrete chloride ion concentration distribution: (**a**) PSSI-200; (**b**) PSSI-400; (**c**) PSSF-50; (**d**) PSSF-100.

**Figure 24 materials-14-06856-f024:**
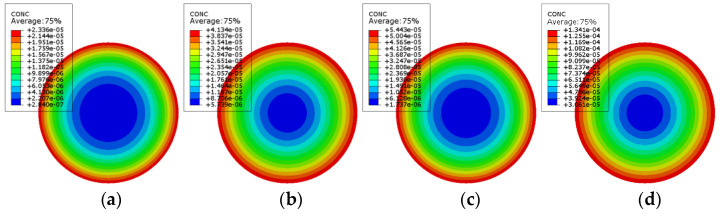
Fully reinforced concrete chloride ion concentration distribution: (**a**) FRSSI-200; (**b**) FRSSI-400; (**c**) FRSSF-50; (**d**) FRSSF-100.

**Figure 25 materials-14-06856-f025:**
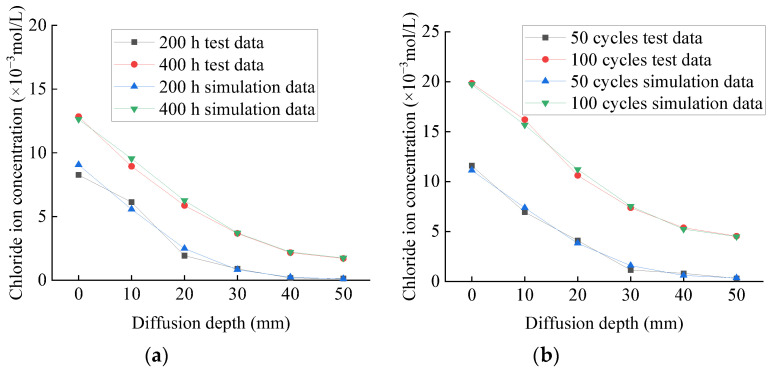
Concentration of chloride ion in plain specimen: (**a**) chlorine salt immersion environment; (**b**) chlorine salt freeze–thaw environment.

**Figure 26 materials-14-06856-f026:**
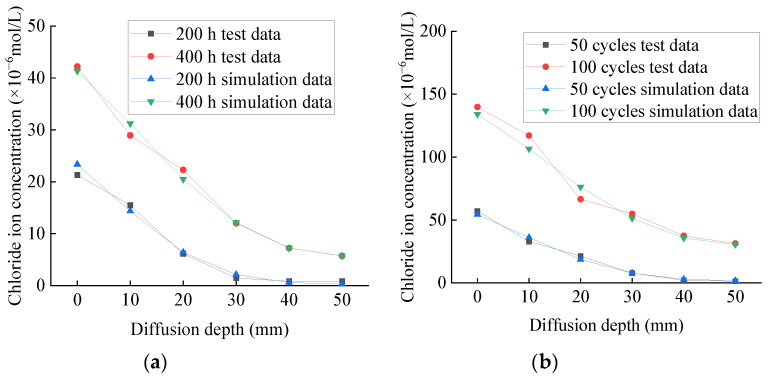
Concentration of chloride ion in fully reinforced specimen: (**a**) chlorine salt immersion environment; (**b**) chlorine salt freeze–thaw environment.

**Figure 27 materials-14-06856-f027:**
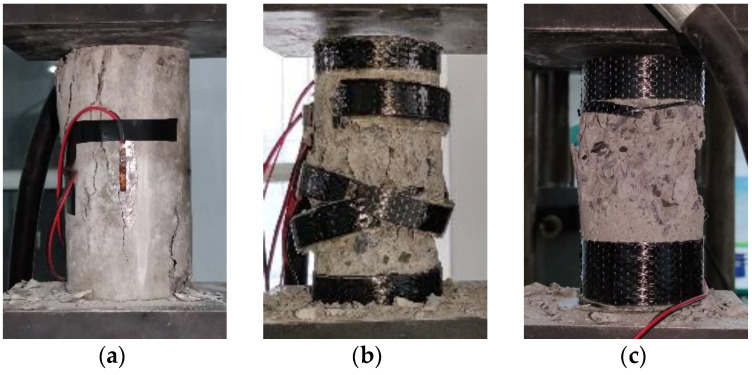
Failure form of specimen: (**a**) plain specimen; (**b**) partially reinforced specimen; (**c**) fully reinforced specimen.

**Figure 28 materials-14-06856-f028:**
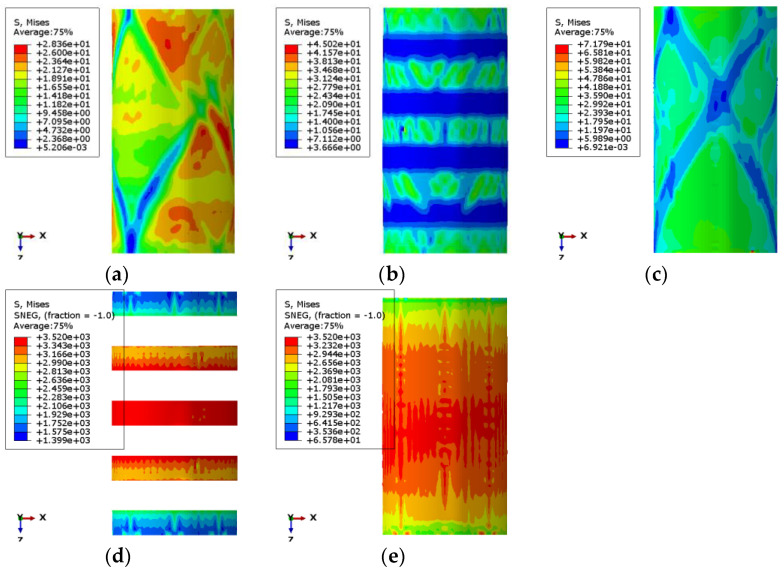
Stress nephogram of finite element model: (**a**) PSND; (**b**) PRSND; (**c**) FRSND; (**d**) CFRP of PRSND; (**e**) CFRP of FRSND.

**Figure 29 materials-14-06856-f029:**
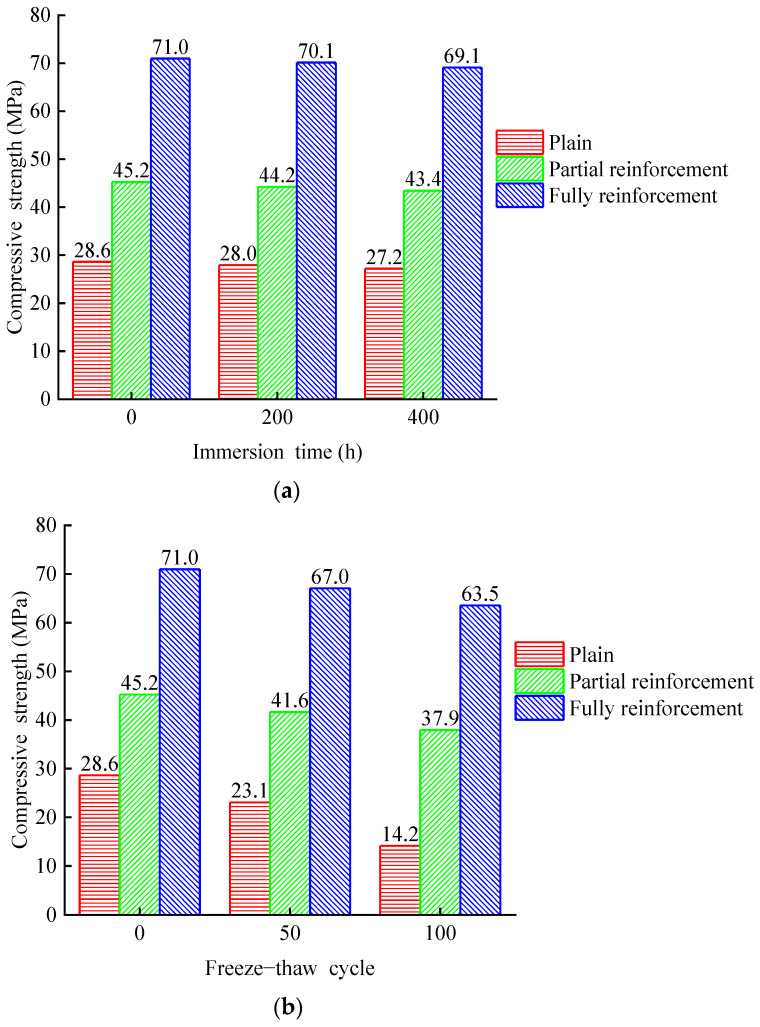
Ultimate compressive strength of specimens: (**a**) chlorine salt immersion environment; (**b**) chlorine salt freeze–thaw environment.

**Figure 30 materials-14-06856-f030:**
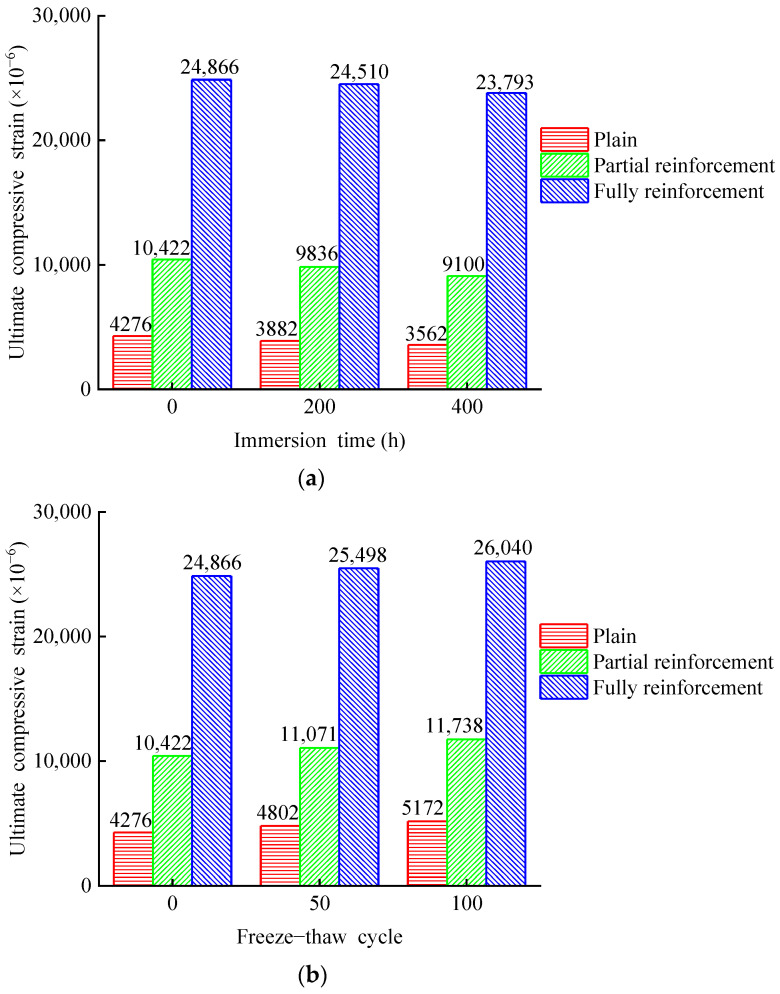
Ultimate strain of specimens: (**a**) chlorine salt immersion environment; (**b**) chlorine salt freeze–thaw environment.

**Figure 31 materials-14-06856-f031:**
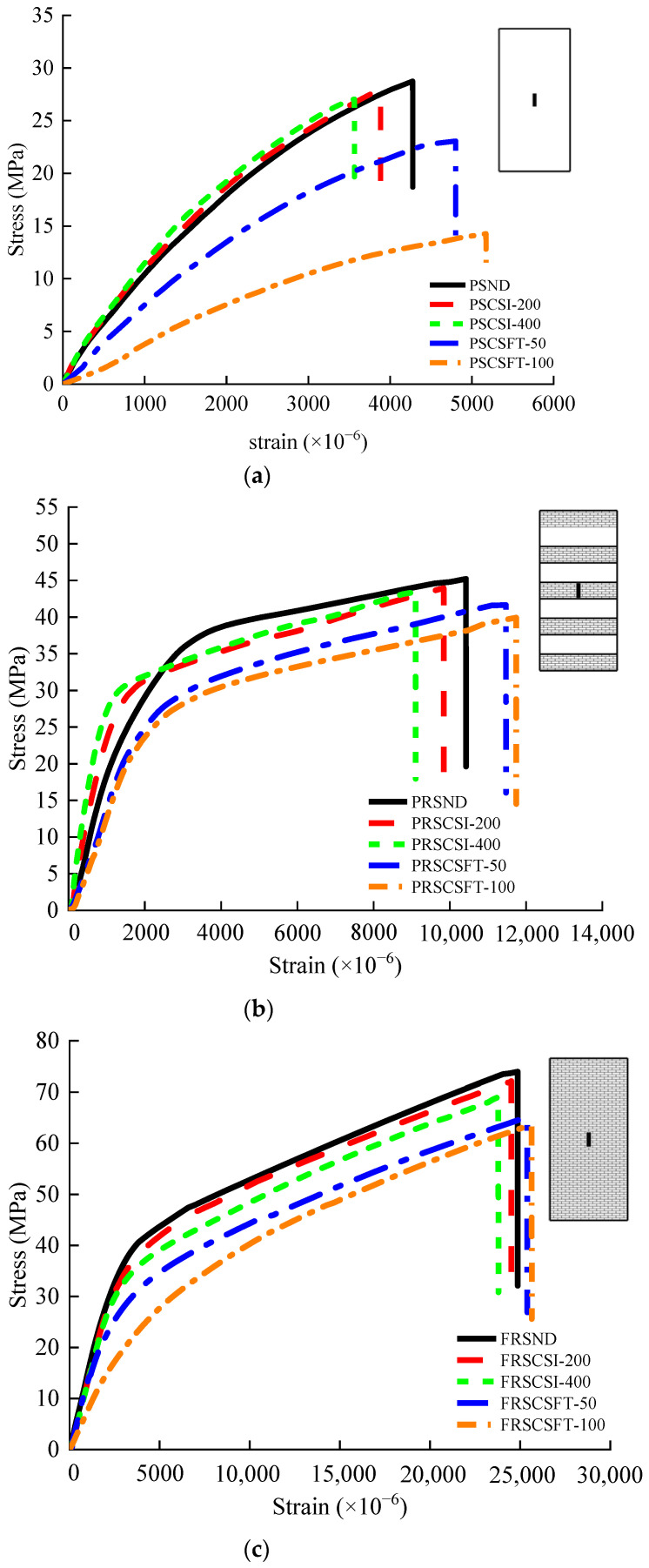
Specimen stress–strain curve: (**a**) plain specimen; (**b**) partially reinforced specimen; (**c**) fully reinforced specimen.

**Table 1 materials-14-06856-t001:** Tensile properties of CFRP and epoxy adhesive.

Material	Tensile Strength/MPa	Elastic Modulus/GPa	Elongation at Break/%
CFRP composite	3520	267	1.78
Epoxy adhesive	54.3	2.7	2.25

**Table 2 materials-14-06856-t002:** Specimens grouping and label information.

	Specimen Type	Plain Specimen	Partially Reinforced Specimen	Fully Reinforced Specimen
Exposure Environment				
No deterioration		PSND	PRSND	FRSND
chlorine salt immersion for 200 h		PSCSI-200	PRSCSI-200	FRSCSI-200
chlorine salt immersion for 400 h		PSCSI-400	PRSCSI-400	FRSCSI-400
chlorine salt freeze–thaw for 50 cycles		PSCSFT-50	PRSCSFT-50	FRSCSFT-50
chlorine salt freeze–thaw for 100 cycles		PSCSFT-100	PRSCSFT-100	FRSCSFT-100

**Table 3 materials-14-06856-t003:** Material properties of concrete.

Compressive Strength of Cube/MPa	Axial Tensile Strength/MPa	Elastic Modulus/MPa
35.36	29.14	35,847.5

**Table 4 materials-14-06856-t004:** Rate of change of ultimate compressive strength and rate of change of ultimate strain of specimens.

Specimen Label	Compressive Strength Variation/%	Compressive Ultimate Strain Variation/%
PSND	—	—
PSCSI-200	−2.35	−9.2
PSCSI-400	−5.03	−16.69
PSCSFT-50	−19.39	12.3
PSCSFT-100	−50.56	20.96
PRSND	—	—
PRSCSI-200	−2.19	−5.63
PRSCSI-400	−3.94	−12.68
PRSCSFT-50	−7.95	6.22
PRSCSFT-100	−16.12	12.63
FRSND	—	—
FRSCSI-200	−1.17	−1.43
FRSCSI-400	−2.65	−4.31
FRSCSFT-50	−5.52	2.54
FRSCSFT-100	−10.47	4.72

## Data Availability

All data generated or analysed during this study are included in this published article.

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
