# Peer review of "Durability Investigation of Carbon Fiber Reinforced Concrete under Salt-Freeze Coupling Effect"

_materials, 2021, doi:10.3390/ma14226856_

Round 1
Reviewer 1 Report
This manuscript presents an interesting experimental about the durability properties of carbon fiber reinforced concrete. The manuscript presentation is clear and contains all the needed experimental details. Also the experimental campaign is well-designed and there is a vast quantity of results. Additionally, the conclusions are well-supported by the results and the discussion. My recommedation is to accept the manuscript since there are no amends proposed to improve the document.
Author Response
Dear Reviewer 1,
The authors would like to thank the reviewers for their review of the manuscript and valuable suggestions. All comments have been carefully considered and accounted for in a revised version of the manuscript. Please review the revised manuscript.
thanks,
Yongcheng Ji

Reviewer 2 Report
The authors propose an experimental and numerical investigation of CFRP reinforced concrete behavior under harsh environment.
In general, the topic is of high interest for many different fields, not only related to civil engineering. The paper is well written, with a clear path to the final conclusions, giving a valuable contribution to the literature. However, there are few issues that need to be clarified in a review.
The introduction is thorough, with good literature review, definition of the motivation and goals of the paper.
In line 37 the meaning of “stable” referring to the chemical property of CFRP is not clear and the statements that their base materials are not reacting with aggressive chemicals is not supported by literature and experience. Please clarify.
In the experimental program a better description of the CFRP material is mandatory. Which kind of material has been used? Which kind of resin? Which kind of texture? A single layer has been used or is a laminated material? Which kind of layup? Is the 0.167 mm the final thickness of the material or the thickness of the single lamina? Is the lamina unidirectional or woven?
All those data are very important for the understanding of the results.
The specimen preparation paragraph is hard to understand, and a better description of the specimens is requested. Line 126 - 127, please clarify the definition of “5 CFRP composites” and “1 CFRP composite”. Line 128 clarify the interface overlap meaning.
The paragraph 2.5 (line 174-177) should be rewritten, and the English checked. Clarify the meaning of “the observation multiple is 550 times”
Same comments for paragraph 2.6, many verbs are missing as in Line 183 and following.
About mechanical test, only the axial strain been recorded. The CFRP seems to be a woven, therefore hoop strain could exist.
About the numerical model, the compression simulation is very complex, and many data are missing in the description. How the plate elements have been connected to the brick elements? How the loads have been applied, and finally how the boundary conditions have been simulated. The results are not clearly described and commented.
Author Response
Dear Reviewer 2,
The authors would like to thank the reviewers for their review of the manuscript and valuable suggestions. All comments have been carefully considered and accounted for in a revised version of the manuscript. Please review the revised manuscript.
thanks,
Yongcheng Ji

Reviewer 3 Report
The presented article describes the tests of cylindrical concrete samples reinforced on the perimeter in the CFRP technology. Tested samples were immersed in corrosive environments.
My main doubt in the scope of the presented tests is the lack of distinction between the properties of samples coated with epoxy resin and samples to which the surface reinforcement was fixed on epoxy resin.
Epoxy resin is a well known and well researched anticorrosive coating of concrete constructions. In the opinion of the reviewer testing of the samples protected with the CFRP on epoxy resin does not make sense and can lead to wrong conclusions. Control samples with epoxy coating only should be tested to point out the differentiation of the tested properties in the field of the research.
In the opinion of the reviewer, due to the incorrectly adopted or described research methodology, the article should not be published in the journal.
Yours Sincerely,
Reviewer.
Author Response
Dear Reviewer 3,
The authors would like to thank the reviewers for their review of the manuscript and valuable suggestions. All comments have been carefully considered and accounted for in a revised version of the manuscript. Please review the revised manuscript.
thanks,
Yongcheng Ji

Reviewer 4 Report
The paper investigates an interesting topic, such as the durability behavior of CFRP (Carbon Fiber Reinforced Polymer) reinforced concrete by considering three category of specimens (plain, partially reinforced, and fully reinforced).
The methodology is pertinent and the structure of the paper well distributed. English is also good. The reviewer suggests some minor modifications: 1. Introduction Express the novelties of the study: how this paper is original? Define the structure of the paper at the end of the paragraph. 2. Paragraphs 3.2 and 3.4 are too long, please divide in sub-paragraphs 3. The numerical model needs to be detailed: 1. how did you validate it? 2. why did you do all the choices? 3. what is the literature that you refer to? 4. Add a paragraph with discussion/results Minor revisions are then requiredAuthor Response
Dear Reviewer 4,
The authors would like to thank the reviewers for their review of the manuscript and valuable suggestions. All comments have been carefully considered and accounted for in a revised version of the manuscript. Please review the revised manuscript.
thanks,
Yongcheng Ji

Round 2
Reviewer 3 Report
Dear Authors
The article could be supplemented with a discussion of the results: their comparison with other works on similar topics cited in the introduction, an indication of the novelties that the article brings to the current state of knowledge and the possibility of practical application of the obtained results.
Yours Sincerelly,
Tomasz Kania.
Author Response
Dear Reviewer,
The authors would like to thank the reviewer for his/her review of the manuscript and valuable suggestions. All comments have been carefully considered and accounted for in a revised version of the manuscript.
Please review it.
Thanks,
Yongcheng Ji
